# Assessment of Arterial Involvement in Pancreatic Cancer: Utility of Reconstructed CT Images Perpendicular to Artery

**DOI:** 10.3390/cancers16122271

**Published:** 2024-06-19

**Authors:** Yoshifumi Noda, Kazuhiro Kobayashi, Masaya Kawaguchi, Tomohiro Ando, Yukiko Takai, Taketo Suto, Yukako Iritani, Takuma Ishihara, Masahiro Fukada, Katsutoshi Murase, Nobuyuki Kawai, Tetsuro Kaga, Toshiharu Miyoshi, Fuminori Hyodo, Hiroki Kato, Tatsuhiko Miyazaki, Nobuhisa Matsuhashi, Kazuhiro Yoshida, Masayuki Matsuo

**Affiliations:** 1Department of Radiology, Gifu University, 1-1 Yanagido, Gifu 501-1194, Japan; kawaguchi.masaya.e1@f.gifu-u.ac.jp (M.K.); ando.tomohiro.y6@f.gifu-u.ac.jp (T.A.); takai.yukiko.f1@f.gifu-u.ac.jp (Y.T.); tkebit9393@gmail.com (T.S.); iritani.rad@gmail.com (Y.I.); kawai.nobuyuki.s9@f.gifu-u.ac.jp (N.K.); kagatetsurow@gmail.com (T.K.); kato.hiroki.w4@f.gifu-u.ac.jp (H.K.); matsuo.masayuki.e0@f.gifu-u.ac.jp (M.M.); 2Department of Frontier Science for Imaging, Gifu University, 1-1 Yanagido, Gifu 501-1194, Japan; 3Department of Pathology, Gifu University Hospital, 1-1 Yanagido, Gifu 501-1194, Japan; kobayashi.kazuhiro.c3@f.gifu-u.ac.jp (K.K.); miyazaki.tatsuhiko.z0@f.gifu-u.ac.jp (T.M.); 4Innovative and Clinical Research Promotion Center, Gifu University Hospital, 1-1 Yanagido, Gifu 501-1194, Japan; ishihara.takuma.d0@f.gifu-u.ac.jp; 5Department of Gastroenterological Surgery and Pediatric Surgery, Graduate School of Medicine, Gifu University, 1-1 Yanagido, Gifu 501-1194, Japan; flyhighvb@yahoo.co.jp (M.F.); murase.katsutoshi.f1@f.gifu-u.ac.jp (K.M.); nobuhisa517@hotmail.com (N.M.); kyoshida@gifu-u.ac.jp (K.Y.); 6Department of Radiology Services, Gifu University Hospital, 1-1 Yanagido, Gifu 501-1194, Japan; miyoshi.toshiharu.n3@f.gifu-u.ac.jp; 7Department of Pharmacology, Graduate School of Medicine, Gifu University, 1-1 Yanagido, Gifu 501-1194, Japan; hyodo.fuminori.i4@f.gifu-u.ac.jp; 8Center for One Medicine Innovative Translational Research (COMIT), Institute for Advanced Study, Gifu University, 1-1 Yanagido, Gifu 501-1194, Japan

**Keywords:** multidetector computed tomography, pancreatic cancer, image processing, computer assisted

## Abstract

**Simple Summary:**

The diagnostic performance and interobserver variability of the contrast-enhanced CT currently used for evaluating arterial involvement from pancreatic cancer has limitations. The purpose of this study was to investigate the utility of reconstructed CT images perpendicular to the artery for assessing arterial involvement from pancreatic cancer and to compare the interobserver variability between it and the current diagnostic imaging method. We found that reconstructed CT images perpendicular to the splenic artery were feasible and showed less interobserver variability than current diagnostic method images.

**Abstract:**

The purpose of this study was to investigate the utility of reconstructed CT images perpendicular to the artery for assessing arterial involvement from pancreatic cancer and compare the interobserver variability between it and the current diagnostic imaging method. This retrospective study included patients with pancreatic cancer in the pancreatic body or tail who underwent preoperative pancreatic protocol CT and distal pancreatectomy. Five radiologists used axial and coronal CT images (current method) and perpendicular reconstructed CT images (proposed method) to determine if the degree of solid soft-tissue contact with the splenic artery was ≤180° or >180°. The generalized estimating equations were used to compare the diagnostic performance of solid soft-tissue contact >180° between the current and proposed methods. Fleiss’ *ĸ* statistics were used to assess interobserver variability. The sensitivity and negative predictive value for diagnosing solid soft-tissue contact >180° were higher (*p* < 0.001 for each) and the specificity (*p* = 0.003) and positive predictive value (*p* = 0.003) were lower in the proposed method than the current method. Interobserver variability was improved in the proposed method compared with the current method (*ĸ* = 0.87 vs. 0.67). Reconstructed CT images perpendicular to the artery showed higher sensitivity and negative predictive value for diagnosing solid soft-tissue contact >180° than the current method and demonstrated improved interobserver variability.

## 1. Introduction

The resectability classification of pancreatic cancer is defined by the National Comprehensive Cancer Network (NCCN) guidelines (Version 2.2024) [1]. According to these guidelines, the presence and degree of solid soft-tissue contact (≤180° or >180°) between the tumor and vessel circumference on pancreatic protocol CT images determines the resectability classification of the tumor as resectable, borderline resectable, or locally advanced [1].

Joo I et al. [2] reported that only 30% of patients with pancreatic cancer were assigned to the same resectability classification by all eight radiologists in their study. Additionally, only fair interobserver agreements in the interpretation of the relationship between the tumor and artery that determines the resectability classification were observed among the radiologists [2]. The researchers stated that the difficulties in differentiating abutment (≤180°) from encasement (>180°) may have caused this interobserver variability [2]. To ensure the objectivity of assessing tumor abutment or encasement in tortuous arteries, reconstructed CT images perpendicular to the arteries can be useful [3]. Indeed, the perpendicular reconstructed CT images demonstrated higher sensitivity (74%) than the axial, coronal, and sagittal images (47–58%) for diagnosing tumor encasement [3].

A discrepancy in interobserver agreements for interpreting CT images may be affected by the observers’ experiences. Previous studies evaluated interobserver variability for arterial involvement on CT images and reported moderate to almost perfect agreement among the reviewers [4,5,6,7]. However, most of the reviewers in their studies were primarily abdominal radiologists or were experienced. One study reported that the more-experienced radiologists (6–10 years of experience in abdominal imaging) showed higher agreement than less-experienced radiologists (first- or second-year fellows) in assessing resectability for arteries (Fleiss’ *ĸ* values of 0.48 vs. 0.19, respectively) [2]. Thus, to improve interobserver variability, a more standardized and universal interpretation method is required. We hypothesized that the use of reconstructed CT images perpendicular to the arteries can achieve high interobserver agreement for assessing arterial involvement from pancreatic cancer. This study aimed to evaluate the utility of perpendicular reconstructed CT images for assessing arterial involvement in patients with pancreatic cancer and to compare the interobserver variability between the conventional image sets (axial and coronal CT images) and perpendicular reconstructed CT images.

## 2. Materials and Methods

### 2.1. Patients

Our institutional review board approved this retrospective study, and written informed consent was waived. Between July 2013 and July 2021, 45 consecutive patients with pancreatic cancer in the pancreatic body or tail who underwent distal pancreatectomy and pancreatic protocol dynamic contrast-enhanced CT before surgery, regardless of the presence or absence of neoadjuvant chemotherapy, were identified. We only included patients with pancreatic cancer in the pancreatic body or tail in this study because comparisons between the pathological findings and reconstructed CT images were easier when perpendicular to the splenic artery than to the celiac or superior mesenteric arteries in cancers in the pancreatic head or uncinate process. Patients who did not have thin-section CT data were excluded (Figure 1).

### 2.2. CT Protocols and Contrast Material Injection

We used a 256- or 64-detector CT scanner (Revolution CT [*n* = 12] or Discovery CT750 HD [*n* = 24]; GE Healthcare, Milwaukee, WI, USA). In Revolution CT scanning, five patients were scanned at 120 kVp in single-energy mode, and seven were scanned in dual-energy mode. Virtual monoenergetic images obtained from dual-energy CT scans were reconstructed at 70 keV. Similarly, in Discovery CT750 HD scanning, eighteen patients were scanned at 120 kVp and six were scanned at 80 kVp, both in single-energy mode. The CT imaging parameters are summarized in Table 1.

The contrast material containing 350 mg iomeprol per mL was intravenously injected over a fixed duration of 30 s. In all patients, 600 mg of iodine per kg of body weight was administered. A circle with a diameter of 15–20 mm was placed as a region of interest in the abdominal aorta at the level of the first lumbar vertebral body. Real-time fluoroscopic monitoring scans (80, 120, or 140 kVp, 10 mA) were initiated 10 s after the start of contrast material injection. The diagnostic CT scanning was started with an additional delay of 20 s for the pancreatic and 60 s for the portal venous phases after a bolus-tracking program (Smart-Prep; GE Healthcare) detected the 100 HU threshold in the abdominal aorta. Coronal reconstructed images of 2.5 mm thickness were available.

### 2.3. Creation of Perpendicular Reconstructed CT Images and Selection of Pathological Specimen

CT data in the pancreatic phase were transferred to a SYNAPSE VINCENT (version 6.8, Fujifilm Co., Ltd., Tokyo, Japan), and reconstructed CT images perpendicular to the splenic artery were created. Using this method, reconstructed CT images could be obtained immediately. The CT images analyzed in this study were obtained before surgery, even in patients treated with neoadjuvant chemotherapy. After loading images on the SYNAPSE VINCENT, a radiologist (M.K., with 9 years of experience) initiated a vascular curved planar reformation analysis application. The radiologist set the start and end points at the proximal and distal portions of the splenic artery with reference to the pancreatic cancer. Then, reconstructed CT images perpendicular to the splenic artery were automatically created.

The same radiologist selected one perpendicular reconstructed CT image that showed the maximum degree of solid soft-tissue contact with the splenic artery, and a pathologist (K.K., with 12 years of experience in tumor pathology) selected a pathological specimen corresponding to the selected perpendicular reconstructed CT image after discussions with the radiologist. They referred to a series of perpendicular reconstructed CT images and pathological specimens and made selections based on the shapes of tumors, splenic arteries and veins, and surrounding structures.

### 2.4. CT Image Analysis

CT image analysis was independently performed by five radiologists, including two board-certified radiologists (T.A. and Y.N., with 9 years of experience in neuroradiology and 11 years of experience in abdominal radiology, respectively [expert]) and three fellows (Y.T., T.S., and Y.I., with 3–5 years of experience [non-expert]) for each patient. The reviewers were aware that all patients had pancreatic cancer; however, they were blinded to the treatment received and pathological findings. We reviewed CT images obtained just before surgery in patients with neoadjuvant chemotherapy. The preset window setting for the axial and coronal CT images was fixed at a 350-HU width and a 40-HU level. The reviewers were allowed to adjust the window settings at their own discretion during the analysis, which consisted of two separate reading sessions with a 2-week interval to minimize recall bias.

During the first-reading session (current method), all reviewers used the axial and coronal CT images to independently determine if the degree of solid soft-tissue contact with the splenic artery was ≤180° or >180°. The radiologist determined the periarterial soft-tissue extended from the tumor to be a solid soft-tissue contact. In the second-reading session (proposed method), all reviewers used the perpendicular reconstructed CT images to independently measure the actual angle of solid soft-tissue contact with the splenic artery and divided the cases into those with ≤180° or >180° on the basis of the measured angles. The angle was measured with reference to the center of the splenic artery.

### 2.5. Reference Standard

The interval between the preoperative pancreatic protocol CT and distal pancreatectomy ranged from 0 to 89 days, with a median of 21 days. A previously mentioned pathologist reviewed all pre-selected pathological specimens. The pathologist separately determined the area of tumor cell infiltration and pancreatic fibrosis (including tumor cell infiltration). Subsequently, the pathologist measured the angles of contact area between tumor cell infiltration and the splenic artery and between pancreatic fibrosis and the splenic artery, respectively, with reference to center of the splenic artery. In the current imaging diagnosis, pathologically confirmed tumor cell infiltration and pancreatic fibrosis are not distinguished on CT images, and we generally recognize pancreatic fibrosis as solid soft-tissue contact on CT images. Therefore, we defined the contact area between the pancreatic fibrosis and splenic artery as a reference standard for the perpendicular reconstructed CT imaging findings.

### 2.6. Statistical Analysis

The sample size was calculated using a Monte Carlo simulation under five fixed reviewers to guarantee the power to detect differences in the accuracy of diagnosis when using the current and proposed methods. First, random numbers were generated from a bivariate Bernoulli distribution for each reviewer, with within-patient correlations set from 0.01 to 0.9, assuming an accuracy of 80% for the current method and 90% for the proposed method. Second, we used generalized estimating equations (GEEs) with patient-level random effects and an exchangeable correlation matrix to test for differences in accuracy between the two images. Finally, the above steps were repeated 1000 times to calculate the minimum sample size for which the empirical power exceeded 80%, which required 36 patients.

Normally distributed variables (patients’ age, body weight, height, and body mass index) were expressed as the mean ± standard deviation (SD) and non-normally distributed variables (CEA level, CA 19–9 level, and tumor size) as the median, along with their interquartile range (IQR). Shapiro–Wilk tests were used to confirm normality. Categorical variables (sex, presence or absence of neoadjuvant chemotherapy, histological type of pancreatic cancer, tumor location, and pathological [p] T stage, pN stage, pM stage, and R classification) were expressed as absolute numbers.

Differences in the pathological and radiological measurements of the solid soft-tissue contacts were evaluated using a pairwise t-test with Bonferroni corrections. As a measure of diagnostic performance, the sensitivity, specificity, positive predictive value (PPV), negative predictive value (NPV), and accuracy were calculated for each method for all reviewers. The GEE model was used for comparisons between the two methods and for the calculation of each measure when the results of all the reviewers were pooled. Fleiss’ kappa was used to evaluate the concordance of the ratings by all reviewers and expert and non-expert radiologists, respectively. Fleiss’ kappa coefficient was evaluated using the benchmark scale proposed by Fleiss [8] on three levels: “<0.4: Poor”, “0.4–0.75: Good”, and “>0.75: Excellent”. For the comparison of concordance between the expert and non-expert radiologists, a z-test was applied under the assumption that the kappa coefficients asymptotically followed a normal distribution. A two-sided *p* value of ≤0.05 was considered to be indicative of statistical significance. The statistical analyses were performed using R software version 4.2.2 (www.r-project.org; accessed on 3 March 2023).

## 3. Results

### 3.1. Patients’ Demographics and Tumor Characteristics

The final sample consisted of 36 patients (21 men and 15 women; mean age ± SD, 73 ± 8 years) (Figure 1). The patients’ demographics and tumor characteristics are summarized in Table 2. Out of the 36 patients, 47% (*n* = 17/36) received preoperative neoadjuvant chemotherapy, and 53% (*n* = 19/36) underwent direct surgical resection. Combined gemcitabine and S–1 (*n* = 9), FOLFIRINOX (a regimen consisting of folinic acid, fluorouracil, irinotecan, and oxaliplatin) (*n* = 6), or combined gemcitabine and nab-paclitaxel (PTX) (*n* = 2) were neoadjuvant chemotherapy options. Pathological responses based on Evans classifications were Grade I (*n* = 2), Grade IIa (*n* = 9), Grade IIb (*n* = 4), Grade III (*n* = 1), and Grade IV (*n* = 1).

### 3.2. Pathological and Radiological Measurements of Solid Soft-Tissue Contact

The measured angles of solid soft-tissue contact with the splenic artery in the pathological specimens and perpendicular reconstructed CT images are summarized in Table 3. Among 36 patients, 17 had pancreatic fibrosis contact >180° and 19 had contact ≤180° (among them, 9 had no contact). The angles of tumor cell infiltration and pancreatic fibrosis in the pathological specimens and solid soft-tissue on perpendicular reconstructed CT images were different in all patients, patients with pancreatic fibrosis contact >180° and ≤180° (*p* < 0.001 for all). In all patients and patients with pancreatic fibrosis ≤180°, the angle of solid soft-tissue on the perpendicular reconstructed CT images was greater than those of tumor cell infiltration and pancreatic fibrosis in the pathological specimens and was greater in pancreatic fibrosis than in tumor cell infiltration (*p* < 0.001 for all). In patients with pancreatic fibrosis >180°, the angle of pancreatic fibrosis was greater than those of tumor cell infiltration and solid soft-tissue on perpendicular reconstructed CT images and was greater in solid soft-tissue on the perpendicular reconstructed CT images than in tumor cell infiltration (Bonferroni-corrected *p* < 0.001 for all pairwise comparisons; Figure 2).

Data are means ± 1 standard deviation, with ranges in parentheses.

### 3.3. Diagnostic Performance in the Current and Proposed Methods

The sensitivity, specificity, PPV, NPV, and accuracy for diagnosing solid soft-tissue contact >180° with pathological results as the reference standard in the current and proposed methods are shown in Table 4. The sensitivity and NPV were higher in the proposed method than in the current method (*p* < 0.001 for both; Figure 3); however, the specificity (*p* = 0.003) and PPV (*p* = 0.03) were lower in the proposed method than in the current method. We detected no difference in the accuracy between the current and proposed methods (*p* = 0.10).

### 3.4. Interobserver Variability

The interobserver variabilities in the arterial involvement assessment among all radiologists, among the expert radiologists, and among the non-expert radiologists, are summarized in Table 5. In the current method, all radiologists showed substantial agreement (*ĸ* = 0.67; 95% confidence interval [CI]: 0.56, 0.77). There was no difference between the expert and non-expert radiologists (*ĸ* = 0.59; 95% CI: 0.28, 0.91 vs. *ĸ* = 0.68; 95% CI: 0.49, 0.87; *p* = 0.32). In the proposed method, there was almost perfect agreement among all radiologists (*ĸ* = 0.87; 95% CI: 0.73, 1.00). There was no difference between the expert and non-expert radiologists (*ĸ* = 1.00; 95% CI: 0.55, 1.00 vs. *ĸ* = 0.79; 95% CI: 0.53, 1.00; *p* = 0.21). On the Fleiss’ benchmark scale, all of the agreements with the proposed method fell into the “Excellent” category.

## 4. Discussion

The degree of solid soft-tissue contact (≤180° or >180°) between pancreatic cancer and the vessel circumference when performing the pancreatic protocol CT determines the resectability classification. However, the diagnostic ability and interobserver variability are low in the current diagnostic imaging method using axial and coronal images [2]. In our study, we initially demonstrated the feasibility of reconstructed CT images perpendicular to the artery for assessing arterial involvement from pancreatic cancer and found that they showed better diagnostic performance and improved interobserver variability than images from the current diagnostic imaging method. A previous study also evaluated the utility of reconstructed CT images perpendicular to the arteries and veins [3]. However, they did not compare the degree of solid soft-tissue contact between the pathological and radiological findings. We distinguished tumor cell infiltration and pancreatic fibrosis, which is related to desmoplastic reaction, in pathological evaluations and compared them with reconstructed CT images perpendicular to the artery. Accordingly, we could discuss the limitations of CT imaging in evaluating arterial involvement, as mentioned below.

Regarding the pathological findings, solid soft-tissue, which we can observe on CT images, is believed to correspond to pancreatic fibrosis rather than tumor cell infiltration in the pathological specimens. Previous studies also have stated that viable tumors and fibrosis are difficult to distinguish on CT images [9,10]. We recognize both tumor cell infiltration and pancreatic fibrosis as solid soft-tissue on CT images. The angle of solid soft-tissue on the perpendicular reconstructed CT images tended to be greater than that of pancreatic fibrosis in the pathological specimens in our study (mean angle: 184° vs. 169°). It is possible that one of the reasons for this finding was the insufficient spatial resolution of CT images. Because this was a retrospective study with a relatively long study period, we could only use images with 1.25 mm thick slices. The spatial resolution could potentially be improved if images with 0.625 mm thick slices were used, which might minimize or eliminate this discrepancy.

The perpendicular reconstructed CT images had a higher sensitivity and NPV than the current method for diagnosing solid soft-tissue contact >180°, with the pathology as the reference standard. On the other hand, the specificity and PPV were lower in the proposed method than in the current method. These results were obtained because the previously described perpendicular reconstructed CT imaging tended to overestimate the contact area of solid soft-tissue compared with the pathological results as the reference standard. However, solid soft-tissue contact on the perpendicular reconstructed CT images could not represent pathological tumor cell infiltration, as described in the previous paragraph. In addition, a recent study demonstrated that solid soft-tissue contact ≤180° evaluated by axial and coronal CT imaging was the best diagnostic criterion for pathologically proven arterial invasion in pancreatic ductal adenocarcinoma [11]. Moreover, the sensitivity for diagnosing arterial invasion was higher for the criterion of solid soft-tissue contact ≤180° (100%) than in solid soft-tissue contact >180° (60%) in patients with neoadjuvant therapy [11]. This finding suggests that even aggressive cases should show solid soft-tissue contact ≤180° on CT images and that the current diagnostic imaging method might underestimate arterial involvement. As shown by these results, there is a definitive limitation to diagnostic imaging in diagnosing arterial invasion. In fact, even though there were 27 patients with at least some solid soft-tissue contact in our study, only three patients had pathologically proven arterial invasion. On the other hand, the NCCN guideline indicates the consideration of neoadjuvant therapy even in resectable cases [1]. As discussed below, we believe that the clinical effect of the significantly improved interobserver variability of the perpendicular reconstructed CT images is higher than the clinical effect of differences in the specificity and PPV. This is because the duration of neoadjuvant therapy is generally different among the resectability classifications [12].

Our study showed almost perfect agreement among all reviewers, and perfect matching was achieved between two expert reviewers. A previous study evaluated the tumor–artery relationship for the celiac axis, superior mesenteric artery, and common hepatic artery [2]. Their study revealed that interobserver agreements for the tumor–artery relationship were fair for all eight reviewers (*ĸ* = 0.33) and the four expert radiologists (*ĸ* = 0.48). Although the evaluated arteries and evaluation methods were different from theirs, our interobserver agreements were far higher than theirs, even for the non-expert radiologists (*ĸ* = 0.79). This finding indicates that perpendicular reconstructed CT imaging can be a more universal diagnostic imaging method to ensure consistency in interobserver variability and treatment strategy.

Our study had several limitations. First, this was a retrospective study conducted in a single center with a relatively small sample size. Therefore, the CT protocols and reconstruction algorithms were heterogeneous. In particular, different kVp settings and image reconstruction techniques were used in this study. This might lead to the difference in image quality. Second, we did not assess the resectability classifications based on the NCCN guidelines and other arteries because we focused on the splenic artery to easily compare the perpendicular reconstructed CT images with the pathological specimen results. Third, we included patients with interval times longer than 30 days from CT scans to distal pancreatectomy. CT findings in these patients might not reflect the pathological results correctly. Indeed, the long interval was associated with tumor progression [13,14]. This might lead to a high PPV and NPV in assessing arterial involvement. Fourth, we did not evaluate venous involvement. For venous involvement, however, the American Journal of Roentgenology expert panel suggests differentiating the evaluations of arterial and venous involvement, because the CT findings differ [15]. The expert panel is suggesting that the degree of solid soft-tissue contact should be used only for arterial involvement. Fifth, we did not assess intraobserver variability. Finally, we could not evaluate the utility of perpendicular reconstructed CT images for diagnosing arterial invasion because only three cases had pathologically proven arterial invasion. Therefore, to validate our results, further clinical studies in evaluating the other arteries, such as celiac, superior mesenteric, and common hepatic arteries, with larger populations are required.

## 5. Conclusions

Reconstructed CT images perpendicular to the artery were feasible for assessing arterial involvement from pancreatic cancer. This method showed improved interobserver variability and higher sensitivity and negative predictive values for diagnosing solid soft-tissue contact >180° than those of the current diagnostic axial and coronal CT imaging methods.

## Figures and Tables

**Figure 1 cancers-16-02271-f001:**
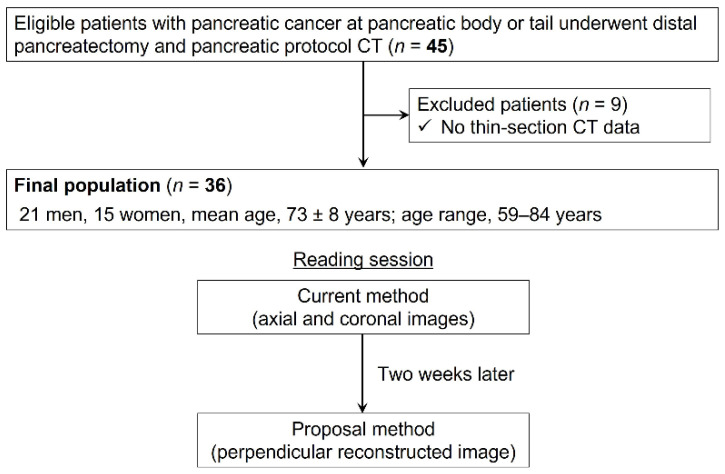
The flowchart shows the numbers of included and excluded patients. Two reading sessions included the current method using axial and coronal CT images and the proposed method using perpendicular reconstructed CT images.

**Figure 2 cancers-16-02271-f002:**
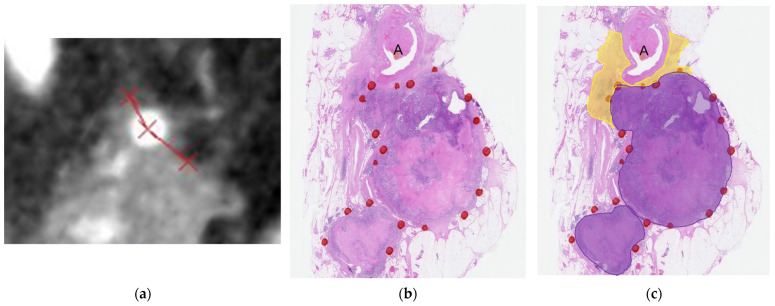
(**a**) Perpendicular reconstructed CT image showing the solid soft-tissue contact with the splenic artery (A). (**b**,**c**) Pathological specimen observed by loupe showing tumor cell infiltration (purple area in (**c**)) and pancreatic fibrosis (yellow area in (**c**)). The angle of the tumor cell infiltration area is 95° and that of pancreatic fibrosis is 188°. The angle of solid soft-tissue contact with the splenic artery on the perpendicular reconstructed CT image is 207°.

**Figure 3 cancers-16-02271-f003:**
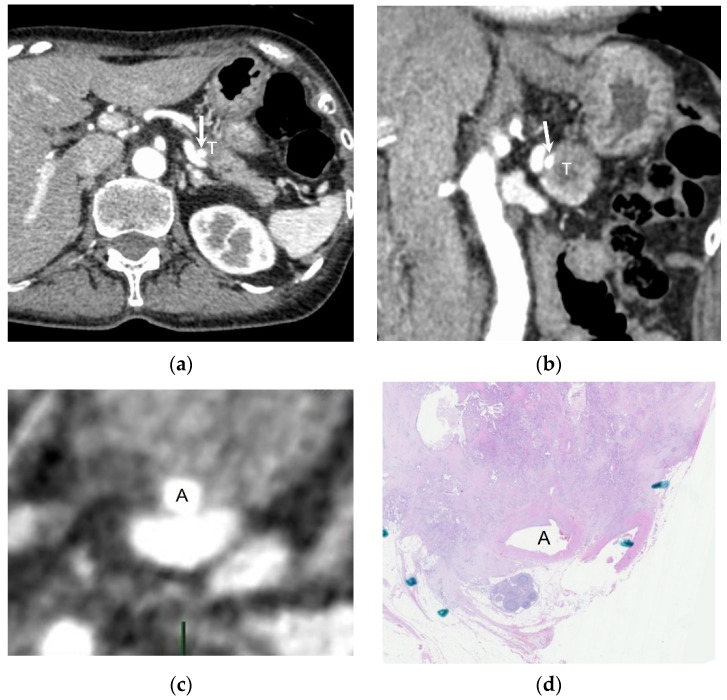
A 71-year-old woman with pancreatic ductal adenocarcinoma at the pancreatic tail. She had not received any systemic chemotherapy. (**a**) Axial and (**b**) coronal CT images showing the tumor (T) as a solid soft-tissue in contact with the splenic artery (arrow). This patient was determined to have solid soft-tissue contact ≤180°. (**c**) Perpendicular reconstructed CT image showing solid soft-tissue contact >180°, and the angle of pancreatic fibrosis is 274° in (**d**) the pathological specimen observed by loupe. A = splenic artery.

**Table 1 cancers-16-02271-t001:** CT imaging parameters.

Parameter	Revolution CT		Discovery CT750 HD	
Scan mode	Single-energy	Dual-energy	Single-energy	Single-energy
Tube voltage (kVp)	120	80/140	120	80
mA modulation	Auto mA and Smart mA	GSI Assist	Auto mA and Smart mA	Auto mA and Smart mA
Noise index (HU)	10.0 HU (at 5 mm slice thickness and ASiR-V 40%)	8.5 HU (at 5 mm slice thickness and ASiR-V 40% at 70 keV)	10.0 HU (at 5 mm slice thickness and FBP)	10.0 HU (at 5 mm slice thickness and FBP)
Detector configuration	128 detectors with 0.625 mm section thickness	128 detectors with 0.625 mm section thickness	64 detectors with 0.625 mm section thickness	64 detectors with 0.625 mm section thickness
Slice thickness/interval (mm)	1.25/0.625	1.25/0.625	1.25/0.625	1.25/0.625
Helical pitch	0.992:1	0.992:1	0.516:1	0.516:1
Rotation time (s)	0.5	0.5	0.4	0.4
Reconstruction	ASiR-V 40%	TFDL-M	ASiR 30%	ASiR 30%

Note. kVp = kilovolt peak, HU = Hounsfield unit, FBP = filtered back projection, ASiR-V = adaptive statistical iterative reconstruction-Veo, TFDL-M = TrueFidelity DL at medium-strength level, ASiR = adaptive statistical iterative reconstruction.

**Table 2 cancers-16-02271-t002:** Patients’ demographics and tumor characteristics.

Characteristics	Study Sample (*n* = 36)
** Patients’ demographics **	
Age (years) *	73 ± 8 (59–84)
Men/Women	21:15
Body weight (kg) *	55 ± 10 (35–74)
Height (cm) *	158 ± 8 (138–176)
Body mass index (kg/m^2^) *	22 ± 3 (16–32)
CEA (ng/mL) ^†^	3 (2–5)
CA 19–9 (U/mL) ^†^	66 (27–311)
Neoadjuvant chemotherapy (+/−)	17/19
** Tumor characteristics **	
Histological type of pancreatic cancer	
Pancreatic ductal adenocarcinoma	34
Adenosquamous carcinoma	1
Anaplastic carcinoma	1
Tumor size (mm) ^†^	23 (19–35)
Tumor location (pancreatic body/tail)	20/16
pT (0/1a/1b/1c/2/3/4)	0/1/1/9/16/7/1
pN (0/1/2)	15/14/7
pM (0/1)	34/2
Pathological arterial invasion (+/−)	3/33
R classification (0/1/2)	30/6/0

Note. CEA = carcinoembryonic antigen, CA = carbohydrate antigen, pT = pathological T, pN = pathological N, pM = pathological M. * Data are means ± 1 standard deviation with ranges in parentheses. ^†^ Data are medians, with interquartile ranges in parentheses.

**Table 3 cancers-16-02271-t003:** Pathological and radiological measurements of solid soft-tissue contact.

Parameter	Pathology		Radiology
	Tumor Cell Infiltration	Pancreatic Fibrosis	Perpendicular Reconstructed CT Images
All patients (°)	145 ± 135	169 ± 139	184 ± 129
Pancreatic fibrosis >180° (°)	258 ± 99	295 ± 70	287 ± 65
Pancreatic fibrosis ≤180° (°)	44 ± 60	57 ± 71	92 ± 99

Note. N.A. = not applicable, No. = number.

**Table 4 cancers-16-02271-t004:** Diagnostic Performance in the Current and Proposed Methods.

Reviewer	Method	Sensitivity	Specificity	PPV	NPV	Accuracy
All reviewers	Current	69% (59/85)	93% (88/95)	89% (59/66)	77% (88/114)	82% (147/180)
	Proposal	98% (83/85)	79% (75/95)	81% (83/103)	97% (75/77)	88% (158/180)
	*p* value	<0.001	0.003	0.03	<0.001	0.10
Expert radiologists						
Reviewer 1	Current	53% (9/17)	90% (17/19)	82% (9/11)	68% (17/25)	72% (26/36)
	Proposal	100% (17/17)	79% (15/19)	81% (17/21)	100% (15/15)	89% (32/36)
Reviewer 2	Current	82% (14/17)	90% (17/19)	88% (14/16)	85% (17/20)	86% (31/36)
	Proposal	100% (17/17)	79% (15/19)	81% (17/21)	100% (15/15)	89% (32/36)
Non-expert radiologists						
Reviewer 3	Current	59% (10/17)	95% (18/19)	91% (10/11)	72% (18/25)	78% (26/36)
	Proposal	94% (16/17)	79% (15/19)	80% (16/20)	94% (15/16)	86% (31/36)
Reviewer 4	Current	88% (15/17)	95% (18/19)	94% (15/16)	90% (18/20)	92% (33/36)
	Proposal	100% (17/17)	79% (15/19)	81% (17/21)	100% (15/15)	89% (32/36)
Reviewer 5	Current	65% (11/17)	95% (18/19)	92% (11/12)	75% (18/24)	91% (29/36)
	Proposal	94% (16/17)	79% (15/19)	80% (16/20)	94% (15/16)	86% (31/36)

Note. PPV = positive predictive value, NPV = negative predictive value.

**Table 5 cancers-16-02271-t005:** Interobserver Variability in the Current and Proposed Methods.

Method	All Reviewers	Expert	Non-Expert	*p* Value
Current (95% CI)	0.67 (0.56, 0.77)	0.59 (0.28, 0.91)	0.68 (0.49, 0.87)	0.32
Proposed (95% CI)	0.87 (0.73, 1.00)	1.00 (0.55, 1.00)	0.79 (0.53, 1.00)	0.21

Note. CI = confidence interval.

## Data Availability

The data presented in this study may be available on request from the corresponding author. The data are not publicly available due to data patient privacy concerns.

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
