# Peer review of "Assessment of Arterial Involvement in Pancreatic Cancer: Utility of Reconstructed CT Images Perpendicular to Artery"

_cancers, 2024, doi:10.3390/cancers16122271_

Round 1
Reviewer 1 Report
Comments and Suggestions for Authors
There are several comments.
It would be better to clarify radiologic features for solid-soft tissue contact with the splenic artery.
It would be better to explain the histologic features of solid-soft tissue.
It would be better to provide detailed and clear instructions on how to measure angles.
It would be better to modify Table 3 for better understanding.
It would be better to add reconstructed CT images perpendicular to the superior mesenteric artery.
Please modify the reference form according to Cancers' guidelines.
Comments on the Quality of English LanguagePlease check English grammar.
For example, Fleiss’s kappa->Fleiss’ kappa
Reviewer 2 Report
Comments and Suggestions for Authors
This is a retrospective study evaluating usefulness of reconstructed CT perpendicular to splenic artery for assessment of artrial involvement of pancreatic cancer.
Major
1. A small sample size is the major limitation. Can the authors add serial CT and corresponding pathological images to increase the sample size? Furthermore, splenic artery involvement does not necessarily affect resectability. Celiac artery or SMA involvement is clinically important. Interobserver agreement can be evalauted for these artery involvement since this does not need surgical specimen.
2. How did the pathologist select the image corresponding to the CT image? The most involved slice might not match between CT and pathological findings.
3. Did the authors evaluate inraobserver agreement, too?
4. Angle of arterial involvement seems to be measured in detail. However, the measurement method was not described both for CT and pathological images. Please describe how the exact angle was measured.
Minor
1. Please add the work load for reconstruction of CT images.
2. Some patients underwent neoadjuvant therapy. Please add data on pathological response. Were CT images evaluated just before surgery in cases with neoadjuvant therapy?
3. Some radiologists argue that perineural invasion cannot be exactly evaluated in very lean patients. Did BMI affect the study results?
Round 2
Reviewer 1 Report
Comments and Suggestions for Authors
The manuscript was well-revised.
It would be better to describe why pancreatic fibrosis occurred (for example, associated with chronic pancreatitis, reactive or tumor stromal change).
Describing reconstructed CT image findings in adenosquamous and anaplastic carcinoma would be better.
Please check whether the Helical Pitch is 0.561 or 0.516.
Comments on the Quality of English LanguagePlease check the tense.
For example, we mainly recognize pancreatic fibrosis as solid -> we mainly recognized pancreatic fibrosis as solid
Reviewer 2 Report
Comments and Suggestions for Authors
The reviewer has no additional comments.
Author Response
We appreciate the reviewer’s comment. Thank you.